# Non-classical amine recognition evolved in a large clade of olfactory receptors

Qian Li[1], Yaw Tachie-Baffour[1], Zhikai Liu[1], Maude W Baldwin[2], Andrew C Kruse[3], Stephen D Liberles[1]*

[1]Department of Cell Biology, Harvard Medical School, Boston, United States; [2]Department of Organismic and Evolutionary Biology, Museum of Comparative Zoology, Harvard University, Cambridge, United States; [3]Department of Biological Chemistry and Molecular Pharmacology, Harvard Medical School, Boston, United States

**Abstract** Biogenic amines are important signaling molecules, and the structural basis for their recognition by G Protein-Coupled Receptors (GPCRs) is well understood. Amines are also potent odors, with some activating olfactory trace amine-associated receptors (TAARs). Here, we report that teleost TAARs evolved a new way to recognize amines in a non-classical orientation. Chemical screens de-orphaned eleven zebrafish TAARs, with agonists including serotonin, histamine, tryptamine, 2-phenylethylamine, putrescine, and agmatine. Receptors from different clades contact ligands through aspartates on transmembrane $\alpha$-helices III (canonical Asp[3.32]) or V (non-canonical Asp[5.42]), and diamine receptors contain both aspartates. Non-classical monoamine recognition evolved in two steps: an ancestral TAAR acquired Asp[5.42], gaining diamine sensitivity, and subsequently lost Asp[3.32]. Through this transformation, the fish olfactory system dramatically expanded its capacity to detect amines, ecologically significant aquatic odors. The evolution of a second, alternative solution for amine detection by olfactory receptors highlights the tremendous structural versatility intrinsic to GPCRs.

*For correspondence: Stephen_liberles@hms.harvard.edu

## Introduction

Biogenic amines are key intercellular signaling molecules that regulate physiology and behavior. In vertebrates, biogenic amines include adrenaline, serotonin, acetylcholine, histamine, and dopamine, and these biogenic amines can be detected by G Protein-Coupled Receptors (GPCRs) expressed on the plasma membrane of target cells. The biochemical and structural basis for biogenic amine recognition by GPCRs is well understood and involves an essential salt bridge between the ligand amino group and a highly conserved aspartic acid on the third transmembrane $\alpha$-helix of the receptor (Asp[3.32]; Ballesteros-Weinstein indexing) (*Katritch et al., 2013*; *Shi and Javitch, 2002*). Mutation of Asp[3.32] in adrenaline, serotonin, acetylcholine, histamine, and dopamine receptors eliminates amine recognition (*Shi and Javitch, 2002*; *Surgand et al., 2006*), and GPCRs that recognize amines through another motif have not been identified.

Trace amine-associated receptors (TAARs) are a family of vertebrate GPCRs distantly related to biogenic amine receptors (*Liberles, 2015*). There are 15 TAARs in mouse, 6 in human, and 112 in zebrafish, and in these or related species, all TAARs except TAAR1 function as olfactory receptors (*Horowitz et al., 2014*; *Hussain et al., 2009*; *Liberles and Buck, 2006*). All human TAARs (6/6), most mouse TAARs (13/15), and some zebrafish TAARs (27/112) retain Asp[3.32], suggesting that many TAARs would recognize amines. High throughput chemical screens yielded ligands for TAAR1 (*Borowsky et al., 2001*; *Bunzow et al., 2001*; *Revel et al., 2011*) and many olfactory TAARs that retain Asp[3.32], including TAAR3, TAAR4, TAAR5, TAAR7s, TAAR8s, TAAR9, and

**eLife digest** Many organisms make molecules called biogenic amines. These molecules, which include the human hormones adrenaline and histamine, have important roles in regulating the biology and behaviour of many animals. Some biogenic amines bind to receptor proteins called GPCRs on the surface of cells. Many drugs can affect the activity of GPCRs, so understanding how different GPCRs work is an important goal of the pharmaceutical industry. Like all proteins, GPCRs are made of chains of molecules called amino acids. The GPCRs that can detect biogenic amines use a particular amino acid named $Asp^{3.32}$, and when this amino acid is mutated, these GPCRs become unable to bind to their target amine.

Trace amine-associated receptors (TAARs) are a type of GPCR that are found in many animals to detect odors. Most TAARs in mammals contain the $Asp^{3.32}$ residue, and recognize amine odors. However, many fish TAARs do not contain $Asp^{3.32}$, and it was not clear what molecules these fish receptors detect.

Here Li et al. find that these fish TAARs also recognize amines, and use a different amino acid called $Asp^{5.42}$. Also, some TAARs contain both $Asp^{3.32}$ and $Asp^{5.42}$, and recognize chemicals with two amines named diamines. Some diamines that bind to TAARs are foul smelling odors; for example, cadaverine and putrescine are repulsive smells emitted by decomposing flesh. In total, the experiments identified amines that can bind to eleven zebrafish TAARs that previously had no odor partner.

Li et al. propose that some fish TAARs lost the $Asp^{3.32}$ during the course of evolution to leave the $Asp^{5.42}$ as the main interaction site for amines. This change dramatically altered how these TAARs interact with amines, which probably expanded the number of different amines that fish can detect. These findings open up new ways to study how the fish brain processes information about its surroundings.

TAAR13c (*Liberles and Buck, 2006*; *Ferrero et al., 2012*; *Hussain et al., 2013*). Each of these receptors detects different amines, some of which evoke innate olfactory behaviors (*Hussain et al., 2013*; *Dewan et al., 2013*; *Ferrero et al., 2011*; *Li et al., 2013*; *Li and Liberles, 2015*). In mouse, TAAR4 detects the aversive carnivore odor 2-phenylethylamine (*Ferrero et al., 2011*), while TAAR5 detects the attractive and sexually dimorphic mouse odor trimethylamine (*Liberles and Buck, 2006*; *Li et al., 2013*). Knockout mice lacking TAAR4 and TAAR5 lose behavioral responses to TAAR ligands identified in cell culture studies (*Dewan et al., 2013*; *Li et al., 2013*).

The selectivity of olfactory TAARs for particular amine odors suggests structural variability within the ligand-binding pocket across the TAAR family. One clade of TAARs (including TAAR1, TAAR3, and TAAR4) detects primary amines derived by amino acid decarboxylation, while a second clade (including TAAR5, TAAR7s, TAAR8s, and TAAR9) prefers tertiary amines (*Ferrero et al., 2012*). Structural models and mutagenesis studies of TAAR1, TAAR7e, and TAAR7f predicted that ligand binding occurs in the membrane plane, $Asp^{3.32}$ forms a salt bridge to the ligand amino group, and other transmembrane residues provide a scaffold that supports ligand binding and enables selectivity (*Ferrero et al., 2012*; *Reese et al., 2014*).

A third TAAR clade (clade III TAARs) arose in fish, and includes the majority (87/112) of zebrafish TAARs (*Hussain et al., 2009*). Most clade III TAARs lack $Asp^{3.32}$ (85/87), and it has been proposed that these receptors may not detect amines (*Hussain et al., 2009*; *Liberles, 2014*). However, ligands have not been identified for any of these receptors. More generally, ligands are known for only one zebrafish TAAR, TAAR13c, which retains $Asp^{3.32}$ and detects the carrion odor cadaverine (*Hussain et al., 2013*). Structure-activity analysis revealed that TAAR13c preferentially recognizes odd-chained, medium-sized diamines, and likely contains two distinct cation recognition sites within the ligand-binding pocket (*Hussain et al., 2013*). However, the structural basis for diamine recognition by TAAR13c was unknown.

Here, we find that TAAR13c recognizes cadaverine through two remote transmembrane aspartic acids, $Asp^{3.32}$ and $Asp^{5.42}$. A TAAR13c variant with a charge-neutralizing mutation at $Asp^{5.42}$ has an altered ligand preference for monoamines rather than diamines. We propose that $Asp^{3.32}$ and $Asp^{5.42}$ together form a general di-cation recognition motif, and note this motif to be present in

several zebrafish, mouse, and human TAARs. Surprisingly, $Asp^{5.42}$ is retained in 84 out of 85 zebrafish TAARs that lack $Asp^{3.32}$, raising the possibility that these receptors recognize amines positioned in a non-canonical inverted orientation with the amino group pointing towards transmembrane α-helix V. Using a high throughput chemical screening approach, we identified ligands for eleven additional zebrafish TAARs, and found that different receptors detect amines through salt bridge contacts at $Asp^{3.32}$ and/or $Asp^{5.42}$. TAAR10a, TAAR10b, TAAR12h, and TAAR12i are $Asp^{3.32}$-containing clade I TAARs that detect serotonin, tryptamine, 2-phenylethylamine, and 3-methoxytyramine, while TAAR16c, TAAR16e, and TAAR16f are $Asp^{5.42}$-containing clade III TAARs that detect N-methylpiperidine, N,N-dimethylcyclohexylamine, and isoamylamine. TAAR13a, TAAR13d, TAAR13e, and TAAR14d have both $Asp^{3.32}$ and $Asp^{5.42}$, and recognize the di-cationic molecules putrescine, histamine, and agmatine.

Based on these findings, we propose that a large clade of olfactory GPCRs evolved a novel way to detect amines that involves a non-classical salt bridge to transmembrane α-helix V. Furthermore, new TAAR ligands include several biogenic amines for which other receptors were identified, including serotonin, histamine, and 2-phenylethylamine. These findings illustrate that different members of the GPCR superfamily can evolve divergent ligand binding pockets for recognition of the same agonist.

## Results

### Two remote amine recognition sites in the cadaverine receptor

To gain structural insights into diamine recognition by the cadaverine receptor TAAR13c, we generated a TAAR13c homology model (*Figure 1A*) based on the X-ray crystal structure of agonist-bound $β_2$ adrenergic receptor (Protein Data Bank Entry 4LDE) (*Ring et al., 2013*). The $β_2$ adrenergic receptor is the closest homolog (33% amino acid identity) of TAAR13c for which an active-state structure is currently available. The homology model of TAAR13c indicated a canonical fold of seven transmembrane α-helices with ligand bound at the core. Using the homology model, we performed computational docking of cadaverine to the orthosteric binding site. One cadaverine amino group of the docked ligand was located near $Asp112^{3.32}$, which corresponds to the highly conserved amine-contact site in biogenic amine receptors. The other cadaverine amino group was positioned at the opposing end of the ligand-binding pocket, 3 Å away from another aspartate, $Asp202^{5.42}$. The second amino group was also within 5 Å of side chains from $Thr203^{5.43}$ and $Ser276^{6.55}$, Leu192 and Phe194 on extracellular loop 2, and the backbone of $Ser199^{5.39}$ (additional nearby residues are depicted in *Figure 1—figure supplement 1*). Based on its proximity and charge, $Asp202^{5.42}$ was deemed a prime candidate to function as a counterion that forms a salt bridge to the second cadaverine amino group. Position 5.42 in several other GPCRs directly contacts ligands through hydrogen bonds, salt bridges, or van der Waals interactions (*Surgand et al., 2006*), and our homology model suggests a role for this amino acid in ligand recognition by TAAR13c.

We used an established reporter gene assay (*Liberles and Buck, 2006*; *Ferrero et al., 2012*) to query the response properties of TAAR13c variants with charge-neutralizing mutations at $Asp^{3.32}$ ($TAAR13c^{D112A}$) and $Asp^{5.42}$ ($TAAR13c^{D202A}$). HEK-293 cells were co-transfected with plasmids encoding TAARs and a cAMP-dependent reporter gene encoding secreted alkaline phosphatase (*Cre-Seap*). Transfected cells were exposed to test ligands, and assayed for reporter activity using a fluorescent phosphatase substrate. As previously published with this paradigm (*Hussain et al., 2013*), cadaverine robustly activated TAAR13c, with a half maximal response occurring at ~10 μM. In contrast, cadaverine failed to activate either the $TAAR13c^{D112A}$ or $TAAR13c^{D202A}$ mutant at any concentration tested, up to 1 mM (*Figure 1*, *Figure 1—figure supplement 1*).

To ensure that mutation of $Asp202^{5.42}$ did not impair protein folding or G protein coupling, we asked whether the mutant receptor could be activated by other ligands. We tested 50 chemicals for the ability to activate $TAAR13c^{D202A}$, and identified the monoamine 3-methoxytyramine as an agonist (*Figure 1D*). 3-methoxytyramine failed to activate wild type TAAR13c at any concentration tested. Thus, charge-neutralizing mutation of $Asp^{5.42}$ transformed TAAR13c from a diamine receptor into a monoamine receptor. The altered ligand-binding preference of the $TAAR13c^{D202A}$ mutant suggests that $Asp^{5.42}$ contributes directly to the ligand-binding pocket, and homology modeling of $TAAR13c^{D202A}$ bound to 3-methoxytyramine (*Figure 1—figure supplement 1*) supports a direct

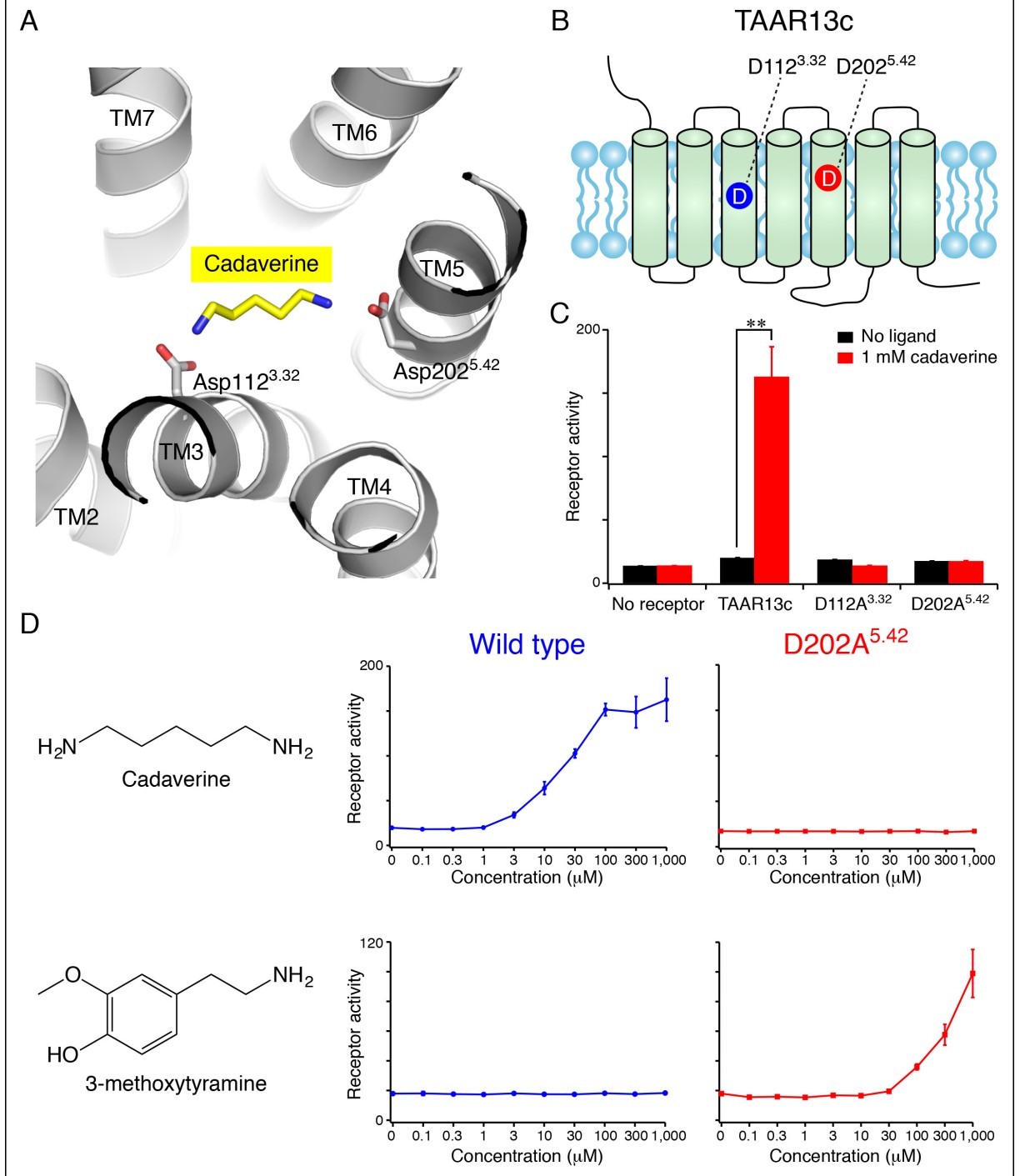

**Figure 1.** Diamine recognition sites in the cadaverine receptor TAAR13c. (**A**) Structural modeling of zebrafish TAAR13c bound to cadaverine (yellow) reveals two aspartates (D112$^{3.32}$ and D202$^{5.42}$) with carboxylates (red) predicted to form salt bridges to ligand amino groups (blue). (**B**) Cartoon of zebrafish TAAR13c topology depicts the location of D112$^{3.32}$ and D202$^{5.42}$. (**C**) Charge-neutralizing mutation of D112$^{3.32}$ and D202$^{5.42}$ abrogates cadaverine responsiveness of zebrafish TAAR13c expressed in HEK-293 cells (mean ± sem, n = 3, **p<0.01 based on two-tailed unpaired Student's *t*-test). (**D**) D202A$^{5.42}$ mutation transforms TAAR13c from a diamine receptor into a monoamine receptor (mean ± sem, n = 3).

The following figure supplements are available for Figure 1:

**Figure supplement 1.** Functional analysis and structural modeling of TAAR13c mutants.

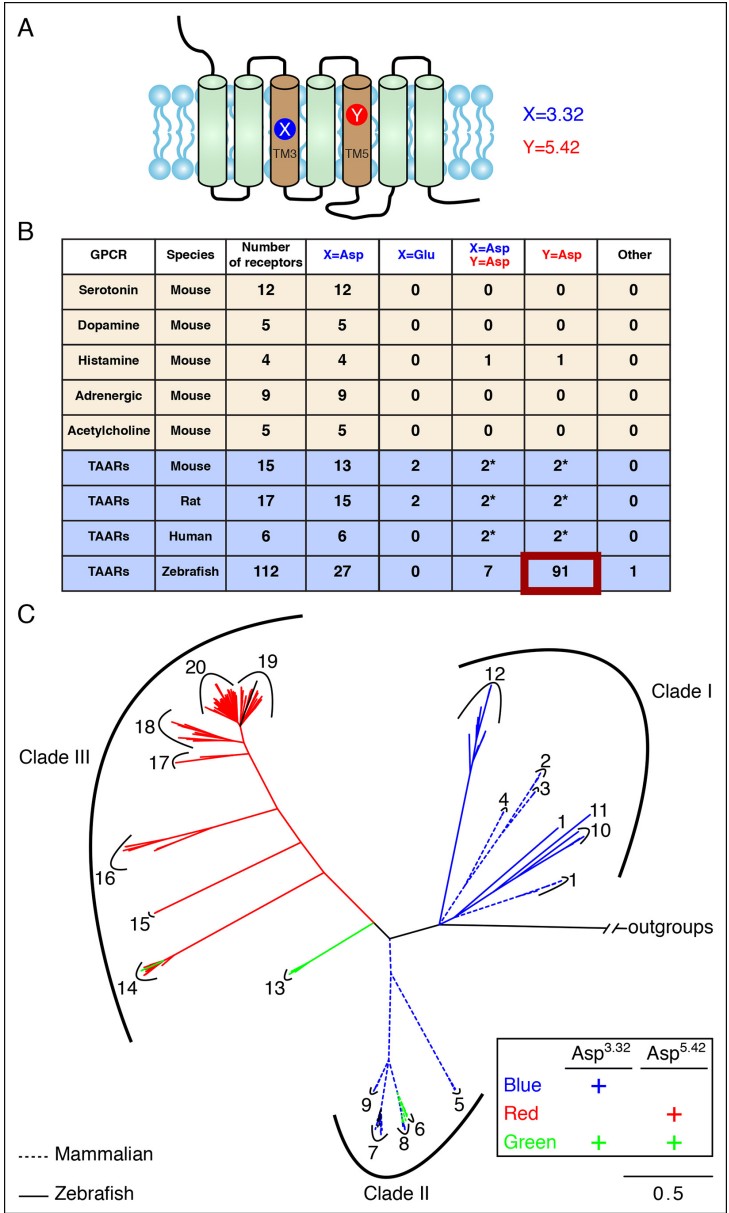

Figure 2. Asp$^{5.42}$ is highly conserved in clade III TAARs. (A) Cartoon depiction indicates the location of positions 3.32 and 5.42 in GPCRs. (B) Bioinformatic analysis reveals the number of receptors with anions at positions 3.32 (X) or 5.42 (Y) across various biogenic receptor subfamilies in mouse, as well as TAARs in mouse, rat, humans, and zebrafish. Two mouse, rat, and human TAARs have Asp$^{5.43}$ instead of Asp$^{5.42}$, and these are included in columns marked Y = Asp (*). (C) Phylogenetic analysis of the TAAR family in zebrafish (solid lines), mice (dashed lines), rats (dashed lines), and humans (dashed lines); scale bar = 0.5 substitutions per site. TAARs containing only Asp$^{3.32}$ (blue), only Asp$^{5.42}$ (red), or both Asp$^{3.32}$ and Asp$^{5.42}$ (green) are depicted. Mammalian TAARs with Asp$^{3.32}$ and Asp$^{5.43}$ instead of Asp$^{5.42}$ are green. Glu$^{3.32}$-containing rodent TAARs and the one zebrafish TAAR, TAAR19f, that lacks both Asp$^{3.32}$ and Asp$^{5.42}$ are depicted in black.

The following figure supplements are available for Figure 2:

**Figure supplement 1.** The phylogenetic tree of tetrapod and fish TAARs.

**Figure supplement 2.** Phylogenetic analysis of TAARs from different fish species.

interaction between this site and ligand. Based on modeling and mutagenesis studies, we conclude that Asp$^{5.42}$ in TAAR13c forms a salt bridge to a cadaverine amino group.

## Asp$^{5.42}$ is widely conserved in clade III TAARs

We next asked whether Asp$^{5.42}$ was present in other GPCRs, and might function more generally in amine detection. Biogenic amine receptors use Asp$^{3.32}$ to contact primary amines (*Shi and Javitch, 2002*; *Surgand et al., 2006*), and Asp$^{5.42}$ is not present in any of 12 serotonin, 5 dopamine, 5 acetylcholine, or 9 adrenergic receptors in mouse (*Figure 2*). Asp$^{3.32}$ is also observed in all 4 mouse histamine receptors, while Asp$^{5.42}$ is observed in one (histamine receptor H2). Histamine is di-cationic at low pH, and the polar imidazole group of the histamine ligand is predicted to contact a transmembrane α-helix 5 asparagine in histamine receptor H1 (Asn$^{5.46}$) and transmembrane α-helix 5 anions in histamine receptors H2 (Asp$^{5.42}$), H3 (Glu$^{5.46}$), and H4 (Glu$^{5.46}$) (*Seifert et al., 2013*).

Most mammalian TAARs retain Asp$^{3.32}$, including 6/6 human TAARs, 13/15 mouse TAARs, and 15/17 rat TAARs. In mouse and rat, two TAAR7s have conservative changes of Asp$^{3.32}$ to Glu$^{3.32}$; the effect of this substitution is unknown as ligands have not been found for Glu$^{3.32}$-containing TAARs. A small number of mammalian TAARs have an aspartate at position 5.43 including two in humans (TAAR6, TAAR8), mice (TAAR6, TAAR8b), and rats (TAAR6, TAAR8a). Sequence alignments indicate that Asp$^{5.43}$ of these receptors corresponds to Asp$^{5.42}$ of TAAR13c, and Asp$^{5.43}$ may be similarly positioned in the agonist-binding pocket. These mammalian TAARs are candidates to function as diamine receptors; however, ligands have not been identified for these receptors, perhaps due to difficulties in achieving their functional expression in heterologous cells. Mice lack a TAAR13c ortholog, yet a role for other TAARs in diamine recognition is supported by the finding that mice lacking all olfactory TAARs fail to avoid cadaverine odor (*Dewan et al., 2013*).

Analysis of the zebrafish TAAR repertoire surprisingly revealed that in contrast to mammalian TAARs and biogenic amine receptors, the vast majority of zebrafish TAARs (91/112) have an aspartate at position 5.42. Some of these receptors (7/91) also retain Asp$^{3.32}$, including TAAR13c, while most (84/91) have lost Asp$^{3.32}$. In several other fish species such as medaka, stickleback, fugu, and salmon, most TAARs also contain Asp$^{5.42}$ but lack Asp$^{3.32}$ (*Figure 2—figure supplement 1*). Phylogenetic analysis indicates that TAARs containing Asp$^{5.42}$ but not Asp$^{3.32}$ are clade III TAARs found in teleosts, and constitute a large branch of the TAAR family tree (*Figure 2*, *Figure 2—figure supplement 2*). Zebrafish lack clade II TAARs, and the teleost TAAR family evolved differently from mammalian TAARs by forming the large cohort of clade III TAARs. Several TAARs with both Asp$^{5.42}$ and Asp$^{3.32}$ reside between clade I TAARs and clade III TAARs phylogenetically (including TAAR13s in zebrafish), suggesting that clade III TAARs derived from clade I TAARs by first gaining Asp$^{5.42}$ and then losing Asp$^{3.32}$.

Ligands for clade III TAARs were unknown, and since they lack the canonical amine recognition site, were proposed to detect chemicals other than amines (*Hussain et al., 2009*; *Liberles, 2014*). However, the widespread retention of Asp$^{5.42}$ in clade III TAARs lacking Asp$^{3.32}$ raised the possibility that the corresponding 84 olfactory receptors in zebrafish indeed detect amines, but do so with ligands positioned in a non-classical orientation.

## Identifying ligands for eleven 'orphan' zebrafish TAARs

We used the high throughput reporter gene assay to identify agonists for additional zebrafish TAARs (*Figure 3*, *Figure 3—figure supplement 1*). Responses of 63 zebrafish TAARs were examined, including at least one representative from each TAAR subfamily in the zebrafish olfactory system (TAAR10 to TAAR20). Zebrafish TAARs were expressed as fusion proteins containing the N-terminal 20 amino acids of bovine rhodopsin, which enhances cell surface expression of some chemosensory receptors in heterologous cells (*Krautwurst et al., 1998*). As above, HEK-293 cells or Hana3A cells (*Saito et al., 2004*) (Hana3A cells express olfactory chaperones for promotion of receptor expression and were used for studies involving TAAR10b, TAAR12i, and TAAR16e) were transfected with plasmids encoding zebrafish TAARs and *Cre-Seap*, incubated with test chemicals, and assayed for reporter phosphatase activity. TAAR ligand identification in heterologous cells has been extensively validated; TAAR ligands identified by this technique also activate TAAR-containing neurons in vivo (*Hussain et al., 2013*; *Dewan et al., 2013*; *Li et al., 2013*; *Zhang, et al., 2013*), and behavioral responses to identified TAAR ligands are lost in TAAR knockout mice (*Dewan et al., 2013*; *Li et al., 2013*). Here, we used the heterologous expression approach to identify the first ligands for zebrafish TAAR10a, TAAR10b, TAAR12h, TAAR12i, TAAR13a, TAAR13d, TAAR13e, TAAR14d, TAAR16c, TAAR16e, and TAAR16f.

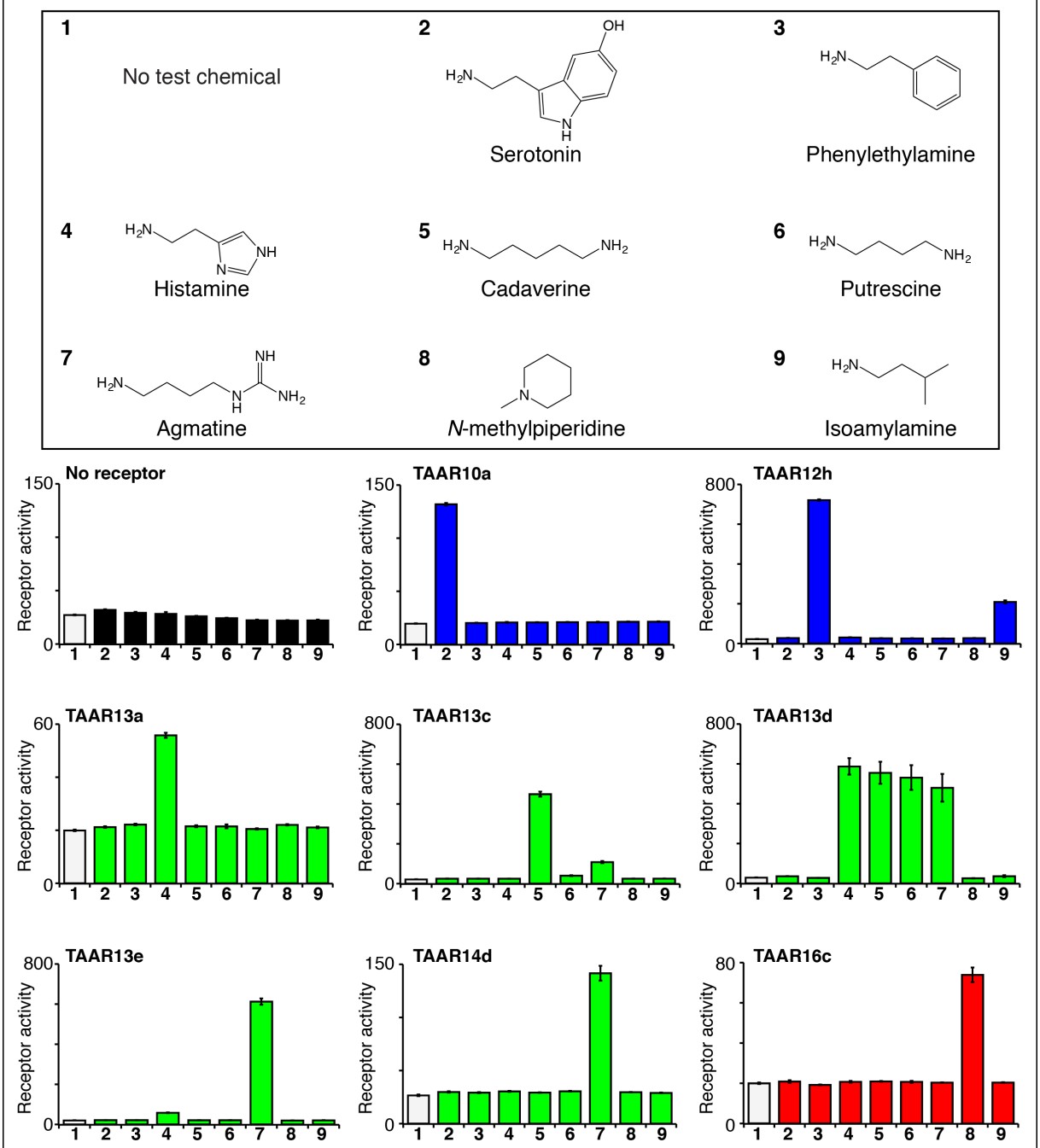

**Figure 3.** Identifying the first ligands for several zebrafish TAARs. HEK-293 cells were cotransfected with *Cre-Seap* and plasmids encoding zebrafish TAARs, incubated with test chemicals (100 µM), and assayed for phosphatase activity using a fluorescent substrate (mean ± sem, n = 3). Zebrafish TAAR10a and TAAR12h are clade I TAARs containing Asp[3.32] but not Asp[5.42] (blue), zebrafish TAAR13a, TAAR13c, TAAR13d, and TAAR13e, and TAAR14d contain both Asp[5.42] and Asp[3.32] (green), and zebrafish TAAR16c is a clade III TAAR containing Asp[5.42] but not Asp[3.32] (red).

The following figure supplements are available for Figure 3:

**Figure supplement 1.** Functional expression of TAAR10b, TAAR12i, and TAAR16e in Hana3A cells.

**Figure supplement 2.** Structure-function studies of zebrafish clade I TAARs: TAAR10a and TAAR12h.

*Ligands for clade I TAARs.* TAAR10a, TAAR10b, TAAR12h, and TAAR12i are clade I TAARs that retain Asp$^{3.32}$ but lack Asp$^{5.42}$. TAAR10a sensitively detects serotonin (EC$_{50}$ = ~0.5 μM), has significantly reduced affinity for the related indole amines 5-methoxytryptamine (EC$_{50}$ = ~20 μM) and tryptamine (EC$_{50}$ = ~70 μM), and does not detect other chemicals examined (*Figure 3—figure supplement 2*). The highest affinity ligand identified for TAAR12h was 2-phenylethylamine (EC$_{50}$ = ~0.3 μM), and TAAR12h also detects other alkyl amines with reduced sensitivity (*Figure 3—figure supplement 2*). TAAR10b and TAAR12i detect the primary amines tryptamine and 3-methoxytyramine respectively (*Figure 3—figure supplement 1*).

Serotonin and 2-phenylethylamine are both biogenic amines that can be recognized by other GPCRs. Zebrafish lack an ortholog of mouse TAAR4, which detects 2-phenylethylamine (*Ferrero et al., 2011*), and mouse TAAR4 shares 41% identity with zebrafish TAAR12h. Likewise, zebrafish TAAR10a is distantly related to mouse serotonin receptors, sharing 27–35% identity. It is thought that the TAAR family derived from an ancestral duplicate of serotonin receptor subtype HTR4 (*Lindemann and Hoener, 2005*), and it is possible that either TAAR10a retains the ligand recognition properties of the ancestral TAAR or that serotonin responsiveness was lost and then re-gained in this lineage.

*Ligands for putative di-cation receptors.* Seven zebrafish TAARs retain both Asp$^{3.32}$ and Asp$^{5.42}$, including all five TAAR13s and two TAAR14s (TAAR14c and TAAR14d). We previously found that zebrafish TAAR13c detects the diamine cadaverine (*Hussain et al., 2013*), and now report that zebrafish TAAR13a, TAAR13d, TAAR13e, and TAAR14d also detect chemicals with two cations.

TAAR13d prefers medium-chained alkyl diamines with highest affinity for putrescine (EC$_{50}$ = ~1 μM), and progressively reduced affinity for diamines with additional methylene groups (*Figure 4A*). TAAR13d has ~3,000-fold reduced sensitivity for diaminopropane, which is only one methylene group shorter than putrescine, indicating a strict minimal agonist length. The differing preference of TAAR13c and TAAR13d for cadaverine and putrescine respectively is consistent with cross-adaptation studies indicating that separate olfactory receptors detect these two diamines (*Rolen, et al., 2003*).

Histamine is an agonist for both TAAR13a (EC$_{50}$ = ~20 μM) and TAAR13d (EC$_{50}$ = ~7 μM). TAAR13d recognizes other alkyl diamines of medium length (see above), while TAAR13a is selective for histamine among tested ligands. TAAR13a and TAAR13d share only 18–28% identity with mouse histamine receptors, and the recognition of histamine by olfactory receptors is likely due to convergent evolution within the GPCR family. Agmatine also activates multiple TAARs, including TAAR13e with higher affinity (EC$_{50}$ = ~1 μM) and TAAR14d with reduced affinity (EC$_{50}$ = ~100 μM). Agmatine is derived from arginine by decarboxylation, reminiscent of several other TAAR ligands that are biogenic amines similarly derived from natural amino acids (*Liberles, 2015*). TAARs that detect agmatine, histamine, serotonin, 2-phenylethylamine, or isoamylamine (see below) are not activated by the natural amino acids from which their ligands were derived (*Figure 4—figure supplement 1*).

*Ligands for clade III TAARs.* Next, we found ligands for three clade III TAARs that lack Asp$^{3.32}$. Despite the absence of the classical amine recognition motif present in all known amine-detecting GPCRs, zebrafish TAAR16c, TAAR16e, and TAAR16f detect amines. Zebrafish TAAR16c detects *N*-methylpiperidine (EC$_{50}$ = ~10 μM), zebrafish TAAR16e detects *N,N*-dimethylcyclohexylamine (EC$_{50}$ = ~30 μM), and zebrafish TAAR16f detects isoamylamine (EC$_{50}$ = ~200 μM).

Removing the amino group of TAAR16c and TAAR16f ligands abolished responses (*Figure 4*, *Figure 4—figure supplement 2*). TAAR16c also detected *N*-methylpyrrolidine (EC$_{50}$ = ~70 μM), and the corresponding secondary amines piperidine and pyrrolidine were partial agonists. In contrast, oxygen-containing analogs, tetrahydropyran and tetrahydrofuran did not activate TAAR16c at any concentration tested, indicating that the amine is essential for receptor agonism. Likewise, TAAR16f failed to detect isoamyl alcohol, and thus TAAR16f agonism also required a ligand amine. These studies indicate that these clade III TAARs are in fact amine receptors, and that the ligand amino group is an essential moiety recognized by the receptor.

## Transmembrane aspartates are required for TAAR agonism

We showed that Asp$^{3.32}$ and Asp$^{5.42}$ are essential for cadaverine detection by TAAR13c (*Figure 1*), and here asked if the corresponding residues are required for activation of other fish TAARs (*Figure 5*). Charge-neutralizing mutation of Asp$^{3.32}$ in clade I TAARs, TAAR10a and TAAR12h, completely eliminated responses to serotonin and 2-phenylethylamine respectively, while charge-

neutralizing mutation of Asp[5.42] in clade III TAARs, TAAR16c and TAAR16f, completely eliminated responses to N-methylpiperidine and isoamylamine respectively. Similarly, mutation of either Asp[3.32]

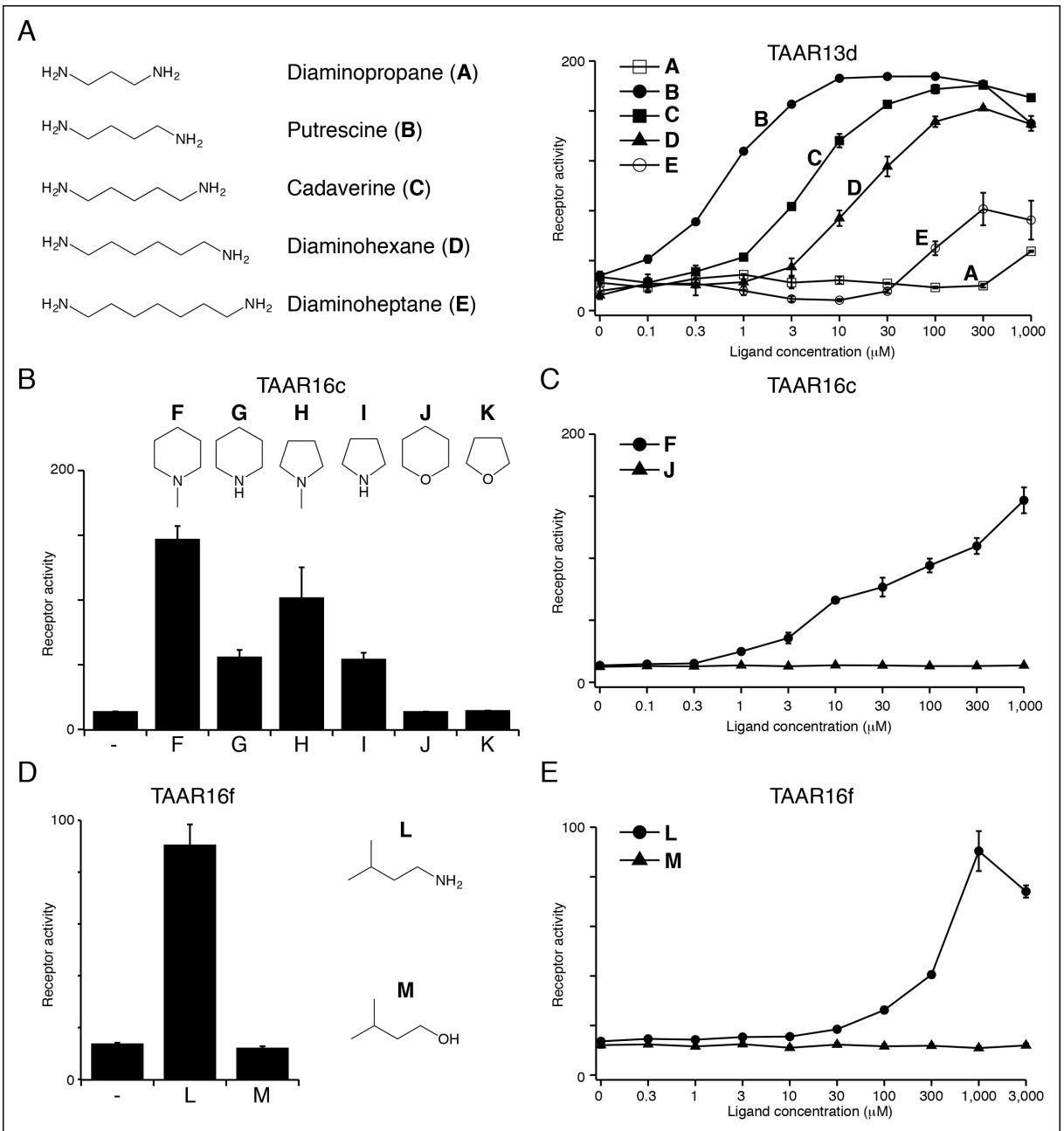

**Figure 4.** Structure-activity studies of zebrafish TAARs. (**A**) Zebrafish TAAR13d displays highest affinity for putrescine among tested ligands, and reduced affinity for longer or shorter diamines. (**B**) Zebrafish TAAR16c recognizes several structurally related amines but not oxygen-containing analogs (1 mM). F: N-methylpiperidine; G: piperidine; H: N-methylpyrrolidine; I: pyrrolidine; J: tetrahydropyran; K: tetrahydrofuran. (**C**) Dose-dependent responses of TAAR16c for N-methylpiperidine and tetrahydropyran. (**D**) Zebrafish TAAR16f recognizes isoamylamine (**L**) but not isoamyl alcohol (**M**) at 1 mM. (**E**) Dose-dependent responses of TAAR16f for isoamylamine and isoamyl alcohol (mean ± sem, n = 3).

The following figure supplements are available for Figure 4:

**Figure supplement 1.** Several TAARs detect amines but not amino acids.

**Figure supplement 2.** Structure-function studies of the zebrafish clade III TAAR, TAAR16c.

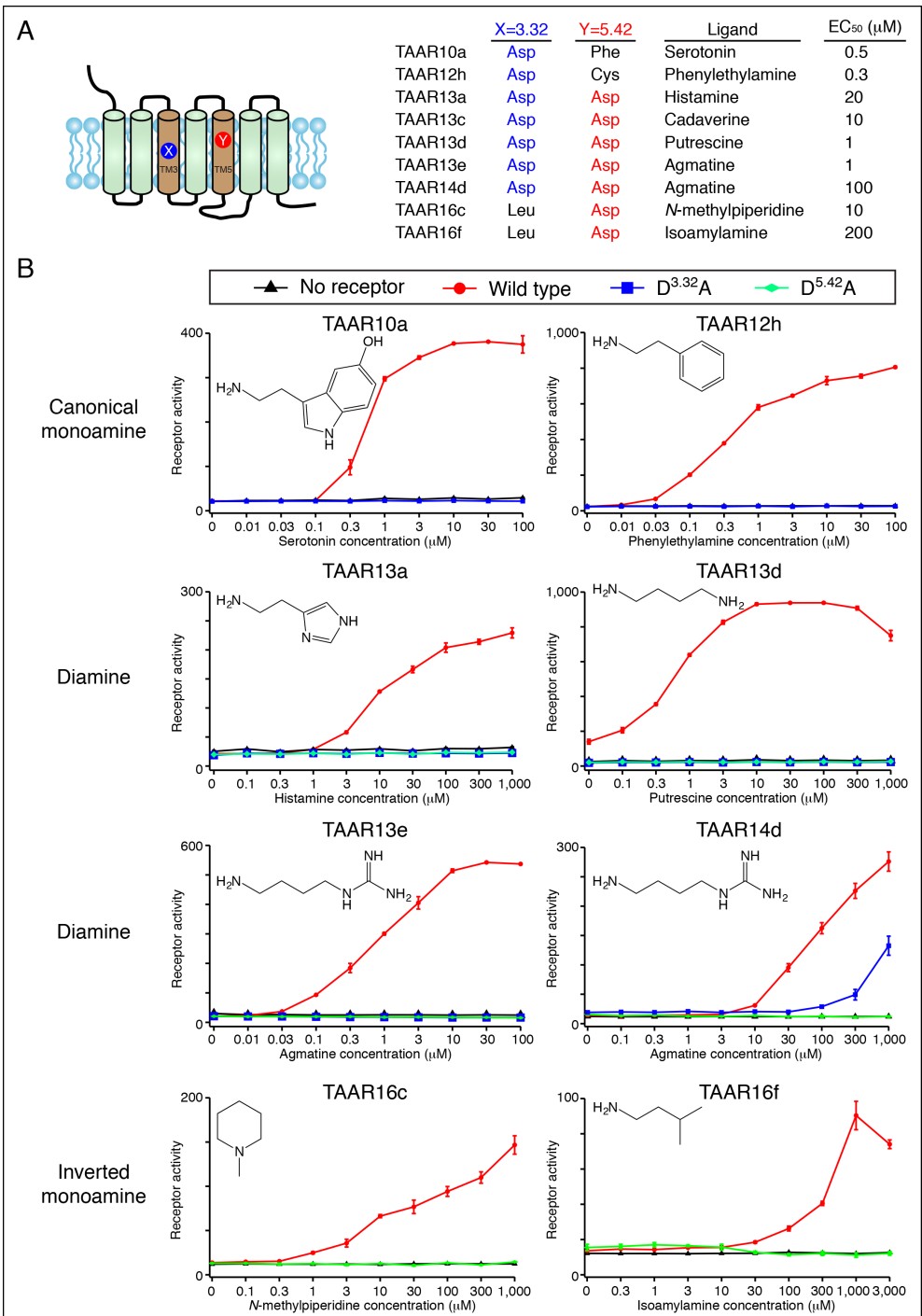

**Figure 5.** Dose-dependent responses of zebrafish TAARs and charge-neutralizing TAAR mutants. (**A**) The identities of amino acids 3.32 and 5.42 in nine 'de-orphaned' zebrafish TAARs, as well as preferred ligands and corresponding $EC_{50}$s, are depicted. (**B**) Dose-dependent activation of TAARs and TAAR mutants by ligands indicated (mean ± sem, n = 3). Responses are shown in cells transfected with *Cre-Seap* alone (black) or together with wild type TAARs (red), $D^{3.32}A$ mutant TAARs (blue), and $D^{5.42}A$ mutant TAARs (green). $D^{3.32}$ is lacking in clade III TAARs (TAAR16c, TAAR16f) and $D^{5.42}$ is lacking in clade I TAARs (TAAR10a, TAAR12h), so the corresponding $D^{3.32}A$ and $D^{5.42}A$ mutants could not be generated.

or $Asp^{5.42}$ in TAAR13a, TAAR13d, TAAR13e, and TAAR14d reduced or abolished responses to all di-cationic ligands. The asymmetric diamines histamine and agmatine could be positioned in either orientation, with the primary amine contacting either $Asp^{3.32}$ or $Asp^{5.42}$. These mutagenesis studies

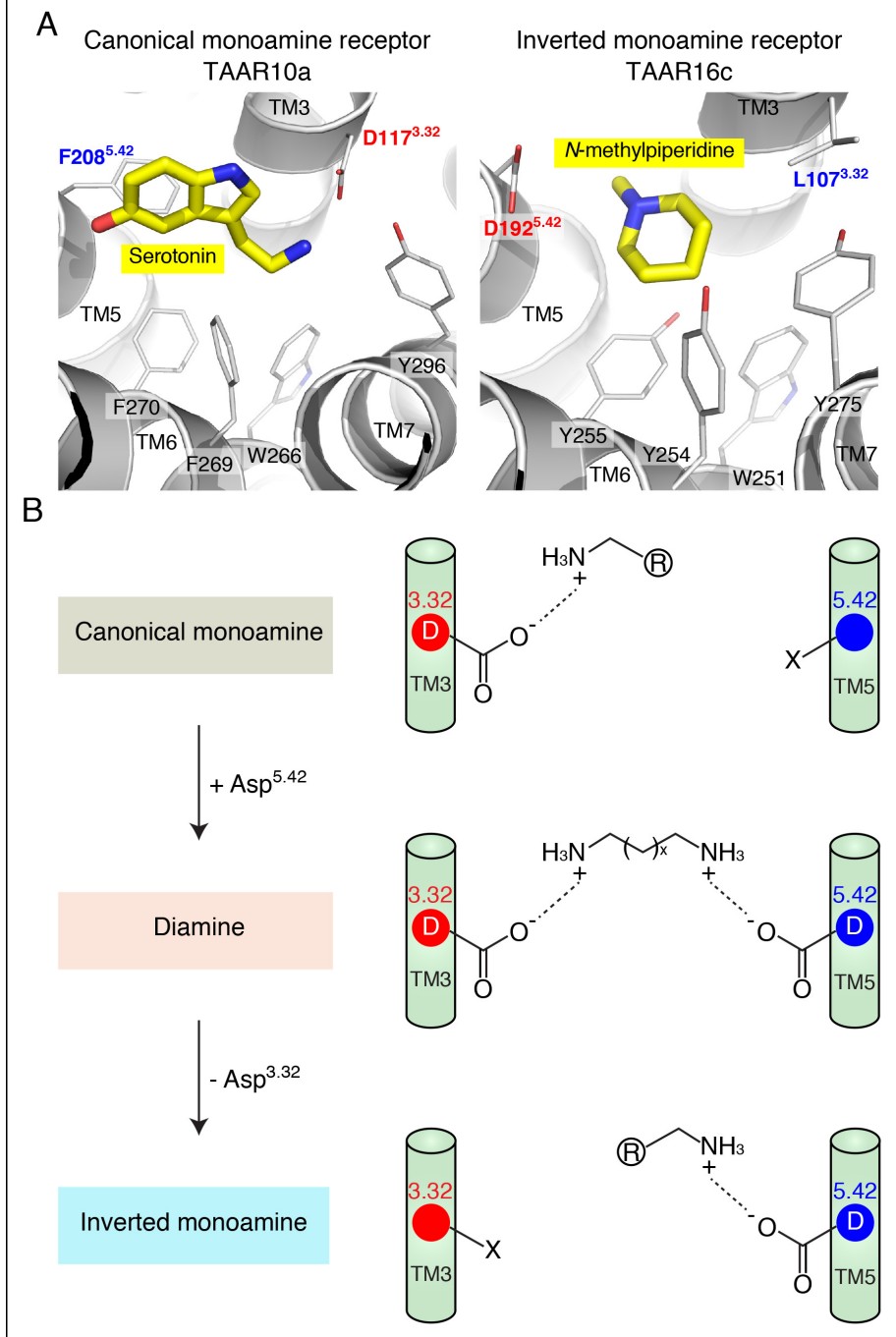

**Figure 6.** Modeling the structure and evolution of non-classical amine recognition. (**A**) Structural modeling of zebrafish TAAR10a and TAAR16c bound to serotonin and *N*-methylpiperidine (yellow) respectively. (**B**) A model for the birth of a large clade of olfactory receptors with non-classical amine recognition. We propose that clade III TAARs evolved a non-classical mode of amine recognition in two steps. First, an ancestral TAAR gained the ability to recognize diamines by acquiring Asp$^{5.42}$. Subsequently, a diamine-detecting TAAR lost the canonical amine recognition site, Asp$^{3.32}$, leading to non-classical amine recognition through a transmembrane α-helix V salt bridge. Extensive gene duplication and mutation expanded and diversified clade III TAARs, leading to a large clade of olfactory receptors with non-canonical amine-detection properties.

The following figure supplements are available for Figure 6:

**Figure supplement 1.** Re-engineering the amine contact site of HTR6.

reveal that clade I TAARs and clade III TAARs require different transmembrane aspartates for amine recognition.

Homology models of serotonin-bound TAAR10a and *N*-methylpiperidine-bound TAAR16c were generated (*Figure 6*), as described previously for cadaverine-bound TAAR13c (*Figure 1A*). TAAR10a is a clade I TAAR and homology modeling indicated proximity of the serotonin amino group to Asp117$^{3.32}$. In contrast, TAAR16c is a clade III TAAR, and homology modeling indicated ligand inversion with the *N*-methylpiperidine amino group near Asp192$^{5.42}$. Inverting the ligand orientation in clade III TAARs would be expected to impact other ligand-receptor contacts; to test this model, we attempted to engineer such a ligand inversion in a canonical biogenic amine receptor, serotonin receptor 6 (HTR6). We analyzed the ligand preference of an HTR6 double mutant (D$^{3.32}$A; A$^{5.42}$D) in which the canonical amine recognition site at Asp$^{3.32}$ was neutralized and a non-canonical amine recognition site at Asp$^{5.42}$ was introduced. Ligand-identification studies revealed that the HTR6 double mutant no longer recognized serotonin, but instead recognized a different amine, ethanolamine (*Figure 6—figure supplement 1*). Re-orienting the ligand amine towards Asp$^{5.42}$ presumably alters other receptor interactions dramatically, allowing Asp$^{5.42}$-containing receptors to sample structurally distinct ligands. Together, these studies indicate that Asp$^{5.42}$ forms a non-classical amine interaction site in GPCRs, and this site is present throughout a large clade of olfactory receptors.

## Discussion

Sensory receptors define the capacity of an organism to perceive its external environment. Olfactory and taste receptor families can rapidly evolve, with dramatic species-specific variations in repertoire size and function (*Nei et al., 2008*; *Shi and Zhang, 2007*). In some lineages, receptor families can be reduced or die out, such as vomeronasal receptors in humans (*Liman, 2006*; *Zhang and Webb, 2003*) or sweet receptors in carnivores (*Jiang et al., 2012*), while in other lineages new receptor families can be born, such as formyl peptide receptors in rodents (*Liberles et al., 2009*; *Rivière et al., 2009*). Functional evolution of chemoreceptor families often involves a pattern of gene duplication and subsequent mutation (*Ferrero et al., 2012*), although isolated functional transformations can also occur, such as taste receptor re-purposing in hummingbirds that led to a novel receptor for sugars found in nectar (*Baldwin et al., 2014*).

Fish are aquatic animals, and efficient detection of water-soluble odors such as amines is important for their ecological niche. Zebrafish generally use the same families of olfactory receptors as other vertebrates, including odorant receptors, vomeronasal receptors, and TAARs (*Kermen et al., 2013*). The TAAR repertoire in mice and humans is relatively small, representing 1–2% of all main olfactory receptors. However, zebrafish have comparable numbers of TAARs (112) and odorant receptors (143), and expansion of the teleost TAAR family suggests a relatively expanded role for TAARs in fish chemosensation. However, prior to this study, ligands were known for only one fish TAAR (*Hussain, et al., 2013*). Moreover, the vast majority of zebrafish TAARs lack the canonical amine recognition motif, and the classes of ligands they might detect were unknown. Here, we show that almost all zebrafish TAARs (111/112) contain either a canonical amine-detection site on transmembrane α-helix III (Asp$^{3.32}$), a non-canonical amine-detection site on transmembrane α-helix V (Asp$^{5.42}$), or both. Thus, teleost TAARs likely represent a very large family of diverse amine receptors.

Several zebrafish TAAR ligands are biogenic amines produced during tissue decomposition (*Shalaby, 1996*). Cadaverine, histamine, agmatine, putrescine, tryptamine, 2-phenylethylamine, and isoamylamine are derived in one step from the natural amino acids lysine, histidine, arginine, ornithine, tryptophan, phenylalanine, and leucine by microbe-mediated decarboxylation. Levels of each of these amines are measured in commercially sold fish as indicators of food quality and freshness (*Shalaby, 1996*); for example, excessive histamine in consumed fish causes human scombroid poisoning (*Morrow et al., 1991*). While some of these biogenic amines are aversive to mammals (*Dewan et al., 2013*; *Ferrero et al., 2011*; *Wisman and Shrira, 2015*), they are reported to evoke variable behaviors in different fish species. For example, agmatine, putrescine, and cadaverine increase feeding-associated pecking behavior in goldfish (*Rolen, et al., 2003*; *Hara, 2006*) while cadaverine and putrescine evoke avoidance behavior in zebrafish (*Hussain et al., 2013*). Future studies are needed to examine the roles of each TAAR and its ligands in fish behavior.

The evolution of divergent solutions to the problem of amine detection by GPCRs reveals the extensive landscape of change possible within the GPCR superfamily. Here, we propose a two-step model for the functional evolution of clade III TAARs that is based on phylogenetic analysis, ligand identification studies, structural modeling, and mutagenesis (*Figure 6*). First, an ancestral clade I TAAR gained Asp$^{5.42}$, and thus the ability to recognize di-cationic chemicals like diamines. Ligands were identified for five zebrafish TAARs that retain Asp$^{3.32}$ and Asp$^{5.42}$, and both aspartates are important for ligand recognition in each receptor. Second, a diamine receptor lost Asp$^{3.32}$, leading to clade III TAARs that display non-classical monoamine recognition through a transmembrane α-helix V salt bridge to Asp$^{5.42}$.

Flipping the amine orientation dramatically changes GPCR-ligand contacts, and presumably allows clade III TAARs to sample a distinct repertoire of structurally divergent amine odors. Once the new amine recognition motif was established, clade III TAARs dramatically expanded through a pattern of gene duplication and subsequent mutation. It seems likely that the massive expansion of this olfactory receptor clade increased the capacity of the fish olfactory system to detect amines, which are water-soluble and ecologically important stimuli for aquatic animals.

## Materials and methods

### Cloning of receptor genes

Zebrafish *Taar* genes were cloned from zebrafish genomic DNA (AB strain), and inserted into a modified pcDNA3.1- (Invitrogen, Waltham, MA, USA) vector containing a 5′ DNA extension encoding the first 20 amino acids of bovine rhodopsin followed by a cloning linker (GCGGCCGCC). Sequencing *Taar* genes revealed several AB strain polymorphisms as compared with Tübingen strain-derived genomic sequences deposited at NCBI. Amino acid-altering polymorphisms identified were as follows: *Taar10a* (NM_001082898): A467G (Ile to Val) and G545A (Val to Ile); *Taar10b* (NM_001082903): A338G (His to Arg), G496A (Ala to Thr), G544A (Val to Ile), and G775A (Val to Met); *Taar12h* (NM_001082909): A23G (Ile to Val), T26C (Tyr to His), C62T (Pro to Ser), C285A (Ser to Tyr), T743A (Phe to Ile), C874T and G875A (Gly to Ile); *Taar12i* (NM_001083085): T23C (Val to Ala), G313T, T314A (Val to Tyr), and T583G (Phe to Val); *Taar13a* (NM_001083102): C427G (Thr to Arg), T680A (Asn to Lys), T803G (Ile to Met), T882C (Phe to Leu), and T948C (Phe to Leu); *Taar13d* (NM_001083041): A112G (Ile to Val), A196C (Thr to Pro), G382A (Val to Ile), and G559A (Val to Ile); *Taar13e* (NM_001083043): A643G (Ile to Val) and G796A (Val to Ile); *Taar16c* (XM_009307036): A797C (Asn to Thr); *Taar16e* (XM_009307037): T103C (Val to Ala), G711A, C712T (Ala to Ile), and C844T (Ala to Val); *Taar16f* (XM_009305637): A23T (Asn to Ile), A59G (Asn to Ser), C190A (His to Lys), A202T (Thr to Ser), and A951C (Leu to Phe). *Htr6* was cloned from mouse brain cDNA and inserted into pcDNA3.1-. *Taar* mutants were generated by overlap extension PCR and the *Htr6* mutant was generated using TagMaster site-directed mutagenesis kit (GM Biosciences, Rockville, MD, USA).

### TAAR functional assays

Reporter gene assays in HEK-293 cells or Hana3A cells (*Saito et al., 2004*) (for studies involving TAAR10b, TAAR12i, and TAAR16e) were performed as previously described (*Liberles and Buck, 2006*; *Ferrero et al., 2012*) with minor modifications. Modifications included use of tissue culture-treated 96 well plates (BD Biosciences, San Jose, CA, USA) pre-incubated with polylysine (250 ng per well); transfection using Lipofectamine 2000 (Invitrogen) reagent according to manufacturer's protocols, and introduction of test stimuli (see below) four to six hours after transfection.

### Test stimuli for high throughput chemical screens

To identify TAAR agonists, chemicals were simultaneously tested in mixtures (83.3 µM per chemical, DMEM). Mix 1 contained GABA, β-alanine, 2-aminopentane, benzylamine, butylamine, and cystamine; mix 2 contained dibutylamine, dimethylamine, *N,N*-dimethylcyclohexylamine, ethylamine, ethyleneamine, and ethanolamine; mix 3 contained hexylamine, isoamylamine, isobutylamine, isopropylamine, indole, and cadaverine; mix 4 contained methylamine, *N*-methylindole, *N*-methylpyrrolidine, *N*-methylpiperidine, pyrrolidine, and putrescine; mix 5 contained hexanal, ethyl butyrate, tryptamine, histamine, 2-phenylethylamine, and trimethylamine; mix 6 contained 2-

methylbutylamine, cysteamine, 1-amino propan-2-ol, 3-methylthiopropylamine, and agmatine; mix 7 contained tyramine, N,N-dimethylethanolamine, octopamine, N,N-dimethylglycine, 5-hydroxyindole-3-acetic acid, and 3-methoxytyramine; mix 8 contained 5-methoxytryptamine, 4-methoxy phenethyl-amine, N,N-dimethylphenethylamine, 5-methoxy-N,N-dimethyltryptamine, N,N-dimethylaniline, N,N-dimethylisopropylamine; mix 9 contained 1-dimethylamino propanol, 2-dimethylaminoethanethiol, and 1-(2-aminoethyl)-pyrrolidine; mix 10 contained 1,7-diaminoheptane, 1,8-diaminooctane, 1,6-dia-minohexane, 1,10-diaminodecane, 1,3-diaminopropane; and mix 11 contained spermine, spermi-dine, dopamine, serotonin, and adenine. TAARs were tested with all 11 chemical mixtures, except TAAR14, TAAR15, and TAAR16 family members which were tested with mixes 1–10.

## TAAR phylogenetic analysis

Amino acid sequences of 112 zebrafish TAARs, 15 mouse TAARs, 17 rat TAARs, 6 human TAARs, 12 biogenic amine receptors (zebrafish HTR2B, HRH2, and DRD2A; mouse HRH3, DRD3, HTR5, DRD1A, and ADRB1; and rat HTR2A, HRH2, DRD3, and ADRB2) and 5 odorant receptors (zebrafish OR103 and OR131, mouse OR121 and OR446, and rat ORI15) were aligned with MAFFT v7.017 (*Katoh, 2002*). For *Figure 2—figure supplements 1, 2*, published TAAR amino acid sequen-ces from opossum, cow, chicken, frog, coelacanth, medaka, stickleback, tetraodon, fugu, Atlantic salmon, and elephant shark, as well as sea lamprey aminergic receptors, were analyzed (*Hussain et al., 2009*; *Tessarolo et al., 2014*). A phylogeny was inferred using a maximum likelihood approach implemented in RAxML (RAxML v7.7.1) using the JTT + $\Gamma$ model of codon sub-stitution, empirical base frequencies, and the rapid bootstrapping technique (100 replicates) (*Stamatakis et al., 2008*). Nodes with support values < 50 were collapsed in the program TreeGraph 2 and trees were visualized using FigTree v1.3.1 (*Stöver and Müller, 2010*).

## TAAR structural modeling

The TAAR13c structural model was generated using SWISS-MODEL workspace (*Biasini et al., 2014*) (http://swissmodel.expasy.org/interactive) and based on the X-ray crystal structure of agonist-bound human $\beta_2$ adrenergic receptor (Protein Data Bank entry 4LDE) (*Ring et al., 2013*). Modeling involved an active-state template, in which agonist binding is favored. The sequence alignment for homology modeling was verified by inspection to confirm proper alignment of sequence landmarks including conserved sequence motifs (DRY/DRH and NPxxY) and transmembrane proline residues. Homology models for TAAR10a (37% amino acid identity) and TAAR16c (30% amino acid identity) were pre-pared similarly using the same template. Models of receptor point mutants were prepared by side chain substitution in previously prepared homology models and then subjected to energy minimiza-tion prior to ligand docking. Docking of ligands into homology models was conducted with the Schrödinger suite (*Schrodinger, 2015*) using LigPrep version 3.4 and Glide version 6.7. First, the receptor model was prepared for docking by adding protons and assigning charges to ionizable side chains. $Asp^{3.32}$ and $Asp^{5.42}$ were modeled in the deprotonated state expected at physiological pH. The resulting model was then subjected to energy minimization using the Schrödinger *impref* utility prior to docking. Ligands were docked to the orthosteric binding pocket using Glide Extra Precision (*Friesner et al., 2006*), and figures were prepared in PyMol (*Schrodinger, 2010*).

## Acknowledgments

We thank Xianchi Dong for assistance with GPCR modeling, Hiroaki Matsunami for Hana3A cells, and Sean Megason for zebrafish genomic DNA. This work was supported by a grant from the NIH (SDL, Award Number R01 DC013289).

## Additional information

### Funding

| Funder | Grant reference number | Author |
| --- | --- | --- |
| National Institute for Health Research | R01 DC013289 | Stephen D Liberles |

The funders had no role in study design, data collection and interpretation, or the decision to submit the work for publication.

## Author contributions

QL, designed and interpreted experiments; wrote the manuscript; cloned TAARs and performed GPCR functional assays; generated and interpreted GPCR homology models; Conception and design, Acquisition of data, Analysis and interpretation of data, Drafting or revising the article; YTB, ZL, designed and interpreted experiments; cloned TAARs and performed GPCR functional assays; Conception and design, Acquisition of data, Analysis and interpretation of data; MWB, designed and interpreted experiments; performed phylogenetic analysis, Conception and design, Acquisition of data, Analysis and interpretation of data; ACK, designed and interpreted experiments;generated and interpreted GPCR homology models; Conception and design, Analysis and interpretation of data; SDL, designed and interpreted experiments; wrote the manuscript; Conception and design, Analysis and interpretation of data, Drafting or revising the article

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
