## [Decision Letter]

Thank you for submitting your manuscript "Non-classical amine recognition evolved in a large clade of olfactory receptors" to *eLife*. The manuscript has been reviewed by a Senior Editor, a Reviewing Editor (Jeremy Nathans), and three expert reviewers, and their assessments, together with my own (Reviewing Editor), form the basis of this letter. I am including the three reviews (lightly edited) at the end of this letter, as there are some specific and useful suggestions in them that will not be repeated in the summary here.

All of the reviewers were impressed with the importance and novelty of your work. The reviewers and I also think that a few additional (and incisive) tests of the model for ligand recognition, using site-directed mutagenesis and functional testing, would greatly strengthen the paper. These are laid out in the last comments of reviewers 2 and 3:

*Reviewer 2:*"First, the description of the homology modeling is a bit too superficial. Additional details should be provided beyond the two highlighted residues (Asp^3.32^ and Asp^5.42^). In addition, it will be important to show the docking of the cadaverine and 3-methoxytyramine in models of both the wild type and mutant receptors. Similarly, can homology modeling of TAAR16c explain the interaction of N-methylpiperidine with the receptor?

Second, the authors could make a stronger case for coordination of cadaverine's second amino group by Asp^5.42^ by studying additional mutations at this position. For example, swapping charges between the receptor and ligand could really bolster their case. In other words, can a D202K or D202R mutant be activated by a cadaverine analog in which one amino group has been replaced by a carboxylate? I recognize that the experiment will fail if the mutation affects receptor expression or receptor folding, or may not be feasible if the compound isn't readily available. The authors might also consider the effects of substitutions of hydrophobic amino acids at D202 that could possibly interact with 1-amino pentane (one positive charge removed from the ligand).

*Reviewer 3:*"To further strengthen the conclusion, you can mutate Asp in TM5 of clade III TAARs (16c or 16f) and introduce Asp back in TM3 and see whether these receptors acquire a ligand binding ability as that of a canonical monoamine receptor type. Also you can mutate Asp in TM3 of 10a or 12h, and in a canonical monoamine receptor, and introduce Asp in TM5, and see whether they bind different ligands like those for the 16c and 16f family."

Full reviews:

*Reviewer #1:*In this manuscript, the authors addressed functional evolution of a family of olfactory receptors, an important and fascinating question given their rapid and dynamic changes in sequences and repertoires throughout animal evolution. They conducted a screening to de-orphan zebrafish TAARs, a major olfactory receptor family in this species, using an established cAMP-mediated reporter gene assay to measure activation in the receptors. They cloned 63 TAARs, about half of the TAARs in the species, and tested against a panel of odorants almost exclusively focusing on amines. They successfully identified active ligands for nine (14%) TAARs, and substantially increased the number of fish TAARs that match with ligands.

The authors conducted nice structure-function analyses focusing on two Asp sites in transmembrane 3 and 5 in combination with structural modeling, and showed that these sites are critical for amine recognition by the tested receptors. Curiously, while the TM3 site is conserved in non-olfactory GPCRs responding to amines, the TM5 site is unique to TAARs. TAAR13 members responding to diamines contain both Asp residues. The authors propose that this new TM5 Asp site expanded the ability to detect amines in fish evolution.

The experiments were rigorously conducted and data analyses were technically solid. However, my concern is that the authors have stretched to discuss evolution of fish TAARs based on the function of a relatively small number of receptors from only one species, which inevitably requires too many assumptions.

Specific points:

1) Despite the authors' screening effort with >60 cloned TAARs, the vast majority of the receptors remain orphan. I wonder if the authors did not test with the right chemicals, or the heterologous expression system used in this study was not efficient (or both). I am concerned if preferred ligands of many class III TAARs are non-amines that the authors did not include for the screening, which would change how to explain Class III evolution. On the other hand, the authors could miss ligands for many TAARs because of difficulty in functional expression in the heterologous system.

2) While I agree with the authors that the de-orphaned receptors would respond to the tested ligand in vivo, what is not clear is how many other receptors would respond to these chemicals, and whether the identified TAARs represent high-affinity receptors for the ligands. These are relevant issues when discussing how evolution shapes specific residues of TAARs.

3) In the third paragraph of the subsection “Asp^5.42^ is widely conserved in clade III TAARs” you state that "TAAR13s represent an evolutionary bridge between clade I and clade III TAARs in zebrafish." It is unclear what "evolutionary bridge" means. Among the fish species the authors show, TAAR13s are unique to zebrafish, raising the concern how TAAR13 members represent as a bridge.

4) I think it is equally possible that the Asp^3.32^/Asp^5.42^ receptor represents the ancestral gene, which would change the order of the proposed TAAR evolution. I'm not sure there's a way to distinguish between these two alternatives (and certainly not with the available data) but the authors need to at least consider different possibilities.

*Reviewer #2:*This is an interesting study that seeks to understand the molecular interactions underlying ligand recognition of the trace amine-associated receptors (TAARs). The TAARs are activated by primary, secondary and tertiary amines and are encoded by a small multigene family in mammals, with most of the family members expressed in olfactory sensory neurons. This particular study takes advantage of an expansion of the TAAR repertoire in zebrafish to investigate the ligand-receptor interactions underlying these receptors' chemical specificities.

The study commences with a homology model of TAAR13c, a receptor that recognizes the diamine cadaverine. Modeling TAAR13c based on the crystal structure of the β_2_ adrenergic receptor, the authors predict direct ionic interactions of the ligand's amino groups with the negatively charged D112 (3.32) in TM3 a D202 (5.42). The interaction of the positively charged amino group with the aspartate in TM5 differentiates this diamine receptor from most other biogenic amine receptors, whose ligands typically contain a single amino group that is coordinated by the aspartate in TM3. The authors then go through a very nice series of experiments to demonstrate the necessity of D202 for activation by cadaverine (D202A mutant), presumably via coordination of one of the positively-charged amino groups since 3-methoxytyramine can activate D202A at high concentrations. These results support the hypothesis that the canonical Asp^3.32^ coordinates the amino groups of monoamine ligands, whereas diamines such as cadaverine can only be accommodated by a second aspartate residue in TM5 (5.42).

Through a phylogenetic analysis of vertebrate TAARs, the authors identify various TAAR receptor containing Asp^3.32^, Asp^5.42^, or both. Interestingly, clade III of the zebrafish TAARs, which represent a large expansion of the family, contain only Asp^5.42^. In support of their model of ligand binding, only receptors possessing both aspartate residues are activated by diamine ligands. Surprisingly, a representative clade III receptor, TAAR16c) can be activated by N-methylpiperidine, a monoamine ligand, and structurally-related compounds. These observations suggest that clade III TAARs bind monoamine ligands in an inverted orientation. The authors propose a model of TAAR evolution in which diamine receptors arose from canonical monoamine receptors via addition of Asp^5.42^, with "inverted" monoamine receptors arising via mutation of Asp^3.32^.

Overall, I find this to be a fascinating story describing the molecular principles of ligand binding in non-canonical biogenic amine receptors with an attractive model of how the TAARs acquire their chemical specificities. I have only a few suggestions that I feel will make an excellent paper even stronger.

First, the description of the homology modeling is a bit too superficial. Additional details should be provided beyond the two highlighted residues (Asp^3.32^ and Asp^5.42^). In addition, it will be important to show the docking of the cadaverine and 3-methoxytyramine in models of both the wild type and mutant receptors. Similarly, can homology modeling of TAAR16c explain the interaction of N-methylpiperidine with the receptor?

Second, the authors could make a stronger case for coordination of cadaverine's second amino group by Asp^5.42^ by studying additional mutations at this position. For example, swapping charges between the receptor and ligand could really bolster their case. In other words, can a D202K or D202R mutant be activated by a cadaverine analog in which one amino group has been replaced by a carboxylate? I recognize that the experiment will fail if the mutation affects receptor expression or receptor folding, or may not be feasible if the compound isn't readily available. The authors might also consider the effects of substitutions of hydrophobic amino acids at D202 that could possibly interact with 1-amino pentane (one positive charge removed from the ligand).

The authors need to show more information from homology modeling.

*Reviewer #3:*To reveal a ligand-binding mode of the trace amine-associated receptor (TAAR) family expressed in the olfactory neurons in fish, Qian et al. identified ligands for previously orphan zebrafish TAARs and mutated either or both of two conserved acidic amino acids (Asp or Glu) in transmembrane domains (TMs) that were predicted to be involved in binding amino groups of ligands based on the knowledge of canonical monoamine receptors. The results of functional analysis together with phylogenetic analysis suggest that the role of Asp in TM3 in recognizing amino groups in canonical monoamine receptors has been taken over by the Asp in TM5 in many of fish TAARs. The functional assays were thoroughly done and the data are convincing.

1) In the Abstract, it is not easy to understand the meaning of 'an unusual 'inverted' orientation'. I do not think you can say it is 'unusual' because it could be said this is usual. In addition, the word 'inverted' may also lead to misunderstanding because it is not clear what is 'right orientation' and what is 'inverted'. Of course, as you read the manuscript, we can understand, but these words are not appropriate to be used in Abstract and thus should be reworded.

2) Despite what the phylogenetic tree suggests, I am not sure what is the experimental evidence to conclude that clade I TAARs in fish first gained Asp in TM5 and then lost Asp in TM3 for clade III TAARs. Also I do not understand why the Asp in TM5 version of TAARs is evolutionally advantageous in fish. Please provide careful discussion.

3) To further strengthen the conclusion, you can mutate Asp in TM5 of clade III TAARs (16c or 16f) and introduce Asp back in TM3 and see whether these receptors acquire a ligand binding ability like that of a canonical monoamine receptor type. Also you can mutate Asp in TM3 of 10a or 12h, and in a canonical monoamine receptor, and introduce Asp in TM5, and see whether they bind different ligands like those for the 16c and 16f family.

---

## [Author Response]

*All of the reviewers were impressed with the importance and novelty of your work. The reviewers and I also think that a few additional (and incisive) tests of the model for ligand recognition, using site-directed mutagenesis and functional testing, would greatly strengthen the paper. These are laid out in the last comments of reviewers 2 and 3:*Reviewer 2:

"First, the description of the homology modeling is a bit too superficial. Additional details should be provided beyond the two highlighted residues (Asp^3.32^ and Asp^5.42^). In addition, it will be important to show the docking of the cadaverine and 3-methoxytyramine in models of both the wild type and mutant receptors. Similarly, can homology modeling of TAAR16c explain the interaction of N-methylpiperidine with the receptor?

Thank you for these comments; we have addressed each point, and think the manuscript is strengthened as a result. First, we provide additional description of the TAAR13c homology model in the main text and methods, and also add an additional supplementary figure that illustrates other predicted ligand-receptor contact sites in the model. Based on this information, it should be evident why Asp^5.42^ was such a prime candidate for subsequent analysis. Second, we provide a homology model for the interaction between 3-methoxytyramine and the TAAR13c D202A^5.42^ mutant. These findings strongly support a direct contribution of Asp^5.42^ to the ligand-binding pocket. Third, we provide additional homology models for TAAR16c (a clade III TAAR) and TAAR10a (a clade I TAAR). In these models, the ligand position in the clade III TAAR is dramatically re-oriented (or inverted) with respect to the ligand position in clade I TAARs or canonical biogenic amine receptors, as the amino group is now directed towards transmembrane α-helix 5.

Second, the authors could make a stronger case for coordination of cadaverine's second amino group by Asp^5.42^ by studying additional mutations at this position. For example, swapping charges between the receptor and ligand could really bolster their case. In other words, can a D202K or D202R mutant be activated by a cadaverine analog in which one amino group has been replaced by a carboxylate? I recognize that the experiment will fail if the mutation affects receptor expression or receptor folding, or may not be feasible if the compound isn't readily available. The authors might also consider the effects of substitutions of hydrophobic amino acids at D202 that could possibly interact with 1-amino pentane (one positive charge removed from the ligand).

This is a really elegant idea, reminiscent of some of the classic 'bump-hole' work on serine protease substrate recognition. For example, it would be exciting if the D202K or D202R mutant recognized amino acids in place of diamines. To test this idea, we generated several D202 mutants: D202K, D202R, D202I, D202L, and D202M, in addition to the D202A mutant originally described. Unfortunately, we were unable to identify ligands for any of the new mutants among our collection of amines and amino acids. These mutants did fail to detect cadaverine, suggesting that the receptors are disrupted in some way. It is possible that there is a potential ligand that we simply lack in our chemical collection. Alternatively, it is possible that these mutations severely impair protein folding, expression, or G protein coupling, and thus prevent functional analysis. While we agree that it would have been interesting to extend these functional studies with a panel of D202 mutants, we feel that the altered ligand specificity of the D202A mutant (together with modeling data) provides compelling evidence that D202 contributes directly to the ligand-binding pocket.

Reviewer 3:

"To further strengthen the conclusion, you can mutate Asp in TM5 of clade III TAARs (16c or 16f) and introduce Asp back in TM3 and see whether these receptors acquire a ligand binding ability as that of a canonical monoamine receptor type. Also you can mutate Asp in TM3 of 10a or 12h, and in a canonical monoamine receptor, and introduce Asp in TM5, and see whether they bind different ligands like those for the 16c and 16f family."

To address this comment, we generated a double mutant of serotonin receptor 6 (HTR6) in which the canonical amine recognition site at Asp^3.32^ was neutralized and a non-canonical amine recognition site at Asp^5.42^ was introduced. Ligand-identification studies revealed that the HTR6 double mutant no longer recognized serotonin, but instead recognized a different amine, ethanolamine. Thus, this double mutation transformed the ligand recognition properties of HTR6, just as the reviewer predicted. We also generated other mutations suggested, but did not identify additional ligand-receptor interactions.

*Full reviews:*Reviewer #1:

Specific points:1) Despite the authors' screening effort with >60 cloned TAARs, the vast majority of the receptors remain orphan. I wonder if the authors did not test with the right chemicals, or the heterologous expression system used in this study was not efficient (or both). I am concerned if preferred ligands of many class III TAARs are non-amines that the authors did not include for the screening, which would change how to explain Class III evolution. On the other hand, the authors could miss ligands for many TAARs because of difficulty in functional expression in the heterologous system.

Heterologous expression of chemosensory receptors has presented major technical difficulties in the field, so while many TAARs remain orphan receptors, we consider identifying the first ligands for 11 TAARs (including the first ligands for any clade III TAARs) an advance. We find that three different clade III TAARs recognize amines (we added new data for a third TAAR in the revision). Importantly, the amine is an essential moiety recognized by TAAR16c and TAAR16f (Figure 4). Removal of the amine group abolishes responses, indicating that amine recognition is absolutely required for agonism of these TAARs. Moreover, we identified a key amine contact site (Asp^5.42^) that based on modeling and mutagenesis studies is critical for ligand recognition in TAAR13c, TAAR16c, and TAAR16f, and in new data, is sufficient to confer amine recognition upon a mutant HTR6 receptor. This amine contact site is preserved in the vast majority of fish TAARs, and based on all of the data presented; we consider it reasonable to propose that these Asp^5.42^-containing TAARs also detect amines.

2) While I agree with the authors that the de-orphaned receptors would respond to the tested ligand in vivo, what is not clear is how many other receptors would respond to these chemicals, and whether the identified TAARs represent high-affinity receptors for the ligands. These are relevant issues when discussing how evolution shapes specific residues of TAARs.

The affinities of several newly de-orphaned TAARs reported here are comparable or better than other olfactory receptors de-orphaned in 293 cells. The best studied mammalian TAARs- TAAR4 and TAAR5- display half maximal responses (EC_50s_) at ligand concentrations of 1-3 μM (2-phenylethylamine) and 300 nM (trimethylamine) respectively. The in vivo relevance for ligand-receptor interactions with this level of affinity is highlighted by studies of TAAR4 and TAAR5 knockout mice, which lose behavioral responses to these odors. The EC_50s_ for several receptor-ligand interactions reported here are comparable to TAAR4 and TAAR5, including TAAR10a (~500 nM serotonin), TAAR12h (~300 nM 2-phenylethylamine), TAAR13d (1 μM putrescine), and TAAR13e (1 μM agmatine). So compared with other olfactory receptors, we consider these to be relatively high affinity interactions. It is certainly possible that other receptors contribute to representations of these odors in vivo, and future studies will examine the roles of particular receptors in neuronal and behavioral responses to these cues.

3) In the third paragraph of the subsection “Asp^5.42^ is widely conserved in clade III TAARs” you state that "TAAR13s represent an evolutionary bridge between clade I and clade III TAARs in zebrafish." It is unclear what "evolutionary bridge" means. Among the fish species the authors show, TAAR13s are unique to zebrafish, raising the concern how TAAR13 members represent as a bridge.

Thank you for this point – we have removed the phrase 'evolutionary bridge'. In several fish species examined, TAARs containing both Asp^5.42^ and Asp^3.32^ reside between clade I TAARs and clade III TAARs on the phylogenetic tree. Based on these findings, we propose that clade III TAARs derived from clade I TAARs by first gaining Asp^5.42^ and then losing Asp^3.32^. Since TAARs lacking both Asp^5.42^ and Asp^3.32^ were not observed at similar locations in the phylogeny, we find no evidence to support the alternative model that clade I TAARs first lost Asp^3.32^ and then gained Asp^5.42^.

4) I think it is equally possible that the Asp^3.32^/Asp^5.42^ receptor represents the ancestral gene, which would change the order of the proposed TAAR evolution. I'm not sure there's a way to distinguish between these two alternatives (and certainly not with the available data) but the authors need to at least consider different possibilities.

There is strong phylogenetic evidence in the literature that TAARs derived from biogenic amine receptors, likely from a serotonin receptor subtype (5-HTR4). Based on this phylogenetic evidence, the ancestral TAAR, like other biogenic amine receptors, contained Asp^3.32^ but not Asp^5.42^.

Reviewer #2:

[…] First, the description of the homology modeling is a bit too superficial. Additional details should be provided beyond the two highlighted residues (Asp^3.32^ and Asp^5.42^). In addition, it will be important to show the docking of the cadaverine and 3-methoxytyramine in models of both the wild type and mutant receptors. Similarly, can homology modeling of TAAR16c explain the interaction of N-methylpiperidine with the receptor?

First, we provide additional description of the TAAR13c homology model in the main text and Methods, and also add an additional supplementary figure that illustrates other predicted ligand-receptor contact sites in the model. Based on this information, it should be evident why Asp^5.42^ was such a prime candidate to form a key salt bridge to the ligand amino group. Second, we provide a homology model for the interaction between 3-methoxytyramine and the TAAR13c D202A mutant. These findings strongly support a direct contribution of Asp^5.42^ to the ligand-binding pocket. Third, we provide additional homology models for TAAR16c (a clade III TAAR) and TAAR10a (a clade I TAAR). In these models, the ligand position in the clade III TAAR is dramatically re-oriented (or inverted) with respect to the ligand position in clade I TAARs or canonical biogenic amine receptors, as the amino group is now directed towards transmembrane α-helix 5.

Second, the authors could make a stronger case for coordination of cadaverine's second amino group by Asp^5.42^ by studying additional mutations at this position. For example, swapping charges between the receptor and ligand could really bolster their case. In other words, can a D202K or D202R mutant be activated by a cadaverine analog in which one amino group has been replaced by a carboxylate? I recognize that the experiment will fail if the mutation affects receptor expression or receptor folding, or may not be feasible if the compound isn't readily available. The authors might also consider the effects of substitutions of hydrophobic amino acids at D202 that could possibly interact with 1-amino pentane (one positive charge removed from the ligand).

This is a really elegant idea, reminiscent of some of the classic 'bump-hole' work on serine protease substrate recognition. For example, it would be exciting if the D202K or D202R mutant recognized amino acids in place of diamines. To test this idea, we generated several D202 mutants: D202K, D202R, D202I, D202L, and D202M, in addition to the D202A mutant originally described. Unfortunately, we were unable to identify ligands for any of the new mutants among our collection of amines and amino acids. These mutants did fail to detect cadaverine, suggesting that the receptors are disrupted in some way. It is possible that there is a potential ligand that we simply lack in our chemical collection. Alternatively, it is possible that these mutations severely impair protein folding, expression, or G protein coupling, and thus prevent functional analysis. While we agree that it would have been interesting to extend these functional studies with a panel of D202 mutants, we feel that the altered ligand specificity of the D202A mutant (together with modeling data) provides compelling evidence that D202 contributes directly to the ligand-binding pocket.

The authors need to show more information from homology modeling.

(Please see above.)

Reviewer #3:

1) In the Abstract, it is not easy to understand the meaning of 'an unusual 'inverted' orientation'. I do not think you can say it is 'unusual' because it could be said this is usual. In addition, the word 'inverted' may also lead to misunderstanding because it is not clear what is 'right orientation' and what is 'inverted'. Of course, as you read the manuscript, we can understand, but these words are not appropriate to be used in Abstract and thus should be reworded.

We removed the phrase 'unusual inverted orientation' in the Abstract and another location in the text.

2) Although the phylogenetic tree suggests, I am not sure what is the experimental evidence to conclude that clade I TAARs in fish first gained Asp in TM5 and then losing Asp in TM3 for clade III TAARs. Also I do not understand why the Asp in TM5 version of TAARs is evolutionally advantageous in fish. Please provide careful discussion.

We discuss the evolutionary advantage in the last paragraph of the conclusion, and added an additional experiment suggested by the reviewer (see below) to strengthen this conclusion. Flipping the amine orientation dramatically changes GPCR-ligand contacts, and presumably offers an evolutionary advantage because it allowed clade III TAARs to sample a distinct repertoire of structurally divergent amine odors. Once the new amine recognition motif was established, this evolutionary innovation was repeatedly harvested as clade III TAARs dramatically expanded through a pattern of gene duplication and subsequent mutation. It seems likely that the massive expansion of this olfactory receptor clade increased the capacity of the fish olfactory system to detect amines, which are water-soluble and ecologically important stimuli for aquatic animals.

The progression of Asp^5.42^ gain then Asp^3.32^ loss is a parsimonious interpretation of the phylogenetic evidence. TAARs containing both Asp^5.42^ and Asp^3.32^ reside between clade I TAARs and clade III TAARs on the phylogenetic tree. Since TAARs lacking both Asp^5.42^ and Asp^3.32^ were not observed at similar locations in the phylogeny, we find no evidence to support the alternative model that clade I TAARs first lost Asp^3.32^ and then gained Asp^5.42^. Furthermore, there is strong phylogenetic evidence in the literature that TAARs derived from biogenic amine receptors, and that the ancestral TAAR, like other biogenic amine receptors, contained Asp^3.32^ but not Asp^5.42^.

3) To further strengthen the conclusion, you can mutate Asp in TM5 of clade III TAARs (16c or 16f) and introduce Asp back in TM3 and see whether these receptors acquire a ligand binding ability as that of a canonical monoamine receptor type. Also you can mutate Asp in TM3 of 10a or 12h, and in a canonical monoamine receptor, and introduce Asp in TM5, and see whether they bind different ligands like those for the 16c and 16f family.

Thank you for this suggestion. We have performed these experiments, and include these new findings in the manuscript. We generated a double mutant of serotonin receptor 6 (HTR6) in which the canonical amine recognition site at Asp^3.32^ was neutralized and a non-canonical amine recognition site at Asp^5.42^ was introduced. Ligand-identification studies revealed that the HTR6 double mutant no longer recognized serotonin, but instead recognized a different amine, ethanolamine. Thus, this double mutation transformed the ligand recognition properties of HTR6, just as the reviewer predicted. We also generated other mutations suggested, but did not identify additional ligand-receptor interactions.